# Generating High-Granularity COVID-19 Territorial Early Alerts Using Emergency Medical Services and Machine Learning

**DOI:** 10.3390/ijerph19159012

**Published:** 2022-07-25

**Authors:** Lorenzo Gianquintieri, Maria Antonia Brovelli, Andrea Pagliosa, Gabriele Dassi, Piero Maria Brambilla, Rodolfo Bonora, Giuseppe Maria Sechi, Enrico Gianluca Caiani

**Affiliations:** 1Electronics, Information and Biomedical Engineering Department, Politecnico di Milano, 20133 Milan, Italy; lorenzo.gianquintieri@polimi.it; 2Civil and Environmental Engineering Department, Politecnico di Milano, 20133 Milan, Italy; maria.brovelli@polimi.it; 3Istituto per il Rilevamento Elettromagnetico dell’Ambiente, Consiglio Nazionale delle Ricerche, 20133 Milan, Italy; 4Azienda Regionale Emergenza Urgenza (AREU), 20124 Milan, Italy; a.pagliosa@areu.lombardia.it (A.P.); g.dassi@areu.lombardia.it (G.D.); p.brambilla@areu.lombardia.it (P.M.B.); r.bonora@areu.lombardia.it (R.B.); g.sechi@areu.lombardia.it (G.M.S.); 5Istituto di Elettronica e di Ingegneria dell’Informazione e delle Telecomunicazioni, Consiglio Nazionale delle Ricerche, 20133 Milan, Italy

**Keywords:** COVID-19, machine learning, health geomatics, geographic information system, emergency medical services, spatial filtering, geo-AI, resources management

## Abstract

The pandemic of COVID-19 has posed unprecedented threats to healthcare systems worldwide. Great efforts were spent to fight the emergency, with the widespread use of cutting-edge technologies, especially big data analytics and AI. In this context, the present study proposes a novel combination of geographical filtering and machine learning (ML) for the development and optimization of a COVID-19 early alert system based on Emergency Medical Services (EMS) data, for the anticipated identification of outbreaks with very high granularity, up to single municipalities. The model, implemented for the region of Lombardy, Italy, showed robust performance, with an overall 80% accuracy in identifying the active spread of the disease. The further post-processing of the output was implemented to classify the territory into five risk classes, resulting in effectively anticipating the demand for interventions by EMS. This model shows state-of-art potentiality for future applications in the early detection of the burden of the impact of COVID-19, or other similar epidemics, on the healthcare system.

## 1. Introduction

Even before the pandemic of COVID-19, the potential central role of data science in infectious disease forecasting and outbreak science was already recognized [1]. In the last two years, the worldwide scientific community has made unprecedented efforts to mitigate the effects of the COVID-19 pandemic, focusing on the application of cutting-edge technologies related to data science: artificial intelligence (AI), machine learning (ML), and big data analytics were considered key assets in extracting information useful to fight the pandemic [2,3,4,5,6,7,8]. Despite the fact that the worst phase of the emergency seems to have passed thanks to the availability of vaccines, it is likely that healthcare systems will have to pay special attention for many years to come to monitor and quickly detect possible events of local recrudescence due to new COVID-19 variants.

Different strategies with three different time horizons have been proposed [3]: (1) the rapid identification of outbreaks and the diagnosis of cases (short-term); (2) the identification of therapeutic options (medium-term); (3) the development of resilient smart cities (long-term). Similarly, the monitoring, surveillance, detection, prevention, and mitigation of indirect effects were identified as goals for applying new technologies, such as big data, AI, the Internet of Things (IoT), and blockchains [7] in public health. In the long term, the use of such tools could represent a unique opportunity to trigger a paradigm shift capable of supporting future policies in public health and medicine.

The spreading of a pandemic is inherently a spatial phenomenon. Coherently, in a recent systematic review [9], a specific focus on different studies that conducted spatial analysis in relation to COVID-19 was explored, in which the following aspects were identified:Spatiotemporal analysis: a descriptive and/or predictive modeling of the evolution of the pandemic within a certain territory (usually at the national and regional level) was explored using official data on positive cases, often also considering people’s mobility, with examples relevant to China [10], South Korea [11], the USA [12], and Italy [13].Health and social geography: the relationship between the virus spreading (based on confirmed cases) and healthcare resources [14], such as nurses [15] or surgeons [16], was explored together with the correlation between confirmed cases and demographic and/or socio-economic characteristics [17,18].Environmental variables: the correlation between confirmed cases and environmental factors, mainly climatic variables [19], such as humidity and temperature [20], was inspected.Data mining: different analyses were performed in relation to additional and alternative data sources, such as mobility [21,22] and flights [23], to corroborate spatial analysis.Web-based mapping: web services implemented to easily visualize and facilitate the comprehension of the obtained results.

From this review, the potentialities of geographic information systems (GIS) as a set of tools for capturing, storing, checking, manipulating, analyzing, and displaying spatially georeferenced data to handle the geospatial component of the pandemic analysis, also at the local level, were highlighted, as in other studies [24,25,26]. In a recent update of this review [27], including 221 papers published in only one year, the importance of data quality, in terms of both availability and spatial-temporal granularity, was underlined, to allow the unveiling of new explicative patterns in the spreading of COVID-19, with higher informative content towards decision-making at the local level. From these reviews, the combination of spatial analysis and data science (mostly AI and specifically ML) has emerged as the best pathway to build descriptive and predictive models to monitor the evolution of the COVID-19 pandemic.

In the recent literature (see Table 1), the implementation of AI predictive models to foresee the evolution of COVID-19 curves in a certain territory has received consistent attention [26,28,29,30,31,32,33], with a specific focus on support for decision-making to implement public health policies [28]. The use of AI methods is recognized as able to outperform the classical statistical models (such as the Susceptible–Exposed–Infectious–Recovered SEIR model) in short-term forecasts [29]. However, some issues still remain unsolved:All the considered models rely on official diagnosis data, which are characterized by significant confounding factors, such as testing capabilities, logistics, data communication flows, and people’s behavior.The geographic resolution of the models remains low, considering whole countries [26,28,29,30,31] or at least large administrative areas [32,33]. A low granularity can consistently limit the effectiveness of models as decision support for policymakers [32].Spatial mapping usually results in a slow generating process [28,31] and therefore is difficult to keep updated and not usable to guide day-by-day activities.

These limitations were addressed in this study by implementing different strategies. Regarding the data source, we previously demonstrated [34] how the geo-localized collection of calls to the emergency medical number and consequent ambulance dispatches by the Emergency Medical Services (EMS) could be considered as an alternative source of information to monitor the epidemic spreading across a territory, as also assessed in previous studies [35,36,37,38,39,40], thus overcoming the intrinsic limitations of the official diagnosis data [8,41,42,43,44], in particular during the first pandemic wave [45]. However, this kind of data was never used, to the best of our knowledge, to implement territorial predictive models. Moreover, EMS data are simple, low in weight, and widely collected, guaranteeing their quick usability, fast processing (thus overcoming the slowness of spatial mapping [28,31]), and high replicability in different territories.

**Table 1 ijerph-19-09012-t001:** Main relevant features of similar studies in the scientific literature.

	Target Variable	Data Source	Max Geographic Granularity	Algorithm Selected	Performance Evaluation
Mollalo et al., 2020 [26]	Cumulative incidence	Socioeconomic, behavioral, environmental, topographic, and demographic factors	County	Multi-Layer Perceptron (MLP)	RMSE = 0.72
Hussein et al., 2022 [28]	Daily infected cases	Official diagnoses	Country	Time-Delay Neural Network (TDNN)	RMSE = 1.15
Alsayed et al., 2020 [29]	Epidemic peak, infected cases	Official diagnoses	Country	Susceptible–Exposed–Infectious–Recovered (SEIR) model, Adaptive Neuro-Fuzzy Inference System (ANFIS)	Normalized RMSE = 0.041
Singh et al., 2020 [30]	Cumulative cases, deaths, recoveries	Official diagnoses	Country	AutoRegressive Integrated Moving Average (ARIMA)	Akaike information criterion value = 20
Hussein et al., 2021 [31]	Daily infected cases	Official diagnoses	Country	Linear forecast model + custom mathematical equation	RMSE = 2.15
Lynch et al., 2021 [32,33]	Cumulative cases	Official diagnoses	County	Moving Average (MA)	MdAE = 0.67
Friedman et al., 2021 [36]	Excess out-of-hospital deaths, respiratory complaints, oxygen saturation level of patients	Emergency Medical Services (EMS) data	City	Comparison against Linear Continuous Fixed Effect	Not applicable
COVID-19 APHP-Universities-INRIA-INSERM Group, 2020 [37]	Requirements for ICU beds	EMS data, positivity ratio, emergency department visits, hospital admissions	Region	Correlation curve analysis	R^2^ = 0.79–0.99
Levy et al., 2021 [38]	Hospitalizations	EMS data	State	AutoRegressive Integrated Moving Average (ARIMA)	AIC
Xie et al., 2021 [40]	EMS demand	Hospitalizations	County	Time series regression	R^2^ = 0.85
Our study	Territorial alert level	EMS data	Municipality	Random Forest (RF)	Accuracy = 80%

Despite the fact that other causes could have contributed to the increase in the number of these events (such as, for example, seasonal flu), the volumes that characterized COVID-19 waves were significantly different [46], so the impact of regular EMS baseline activity for other respiratory or infective issues could therefore be neglected. This kind of approach could also be important in the upcoming scenario, in which we are witnessing progressively decreasing preventive measures towards virus diffusion, which may result in a lower level of population screening and the abrupt local spreading of the disease.

With relevance to the geographic granularity, the limit to overcome is the low statistical meaningfulness of models focusing on small areas in terms of resident population, so that a certain level of aggregation is necessary to identify actual patterns in the data rather than random noise. To solve this issue, a new custom method for spatial aggregation is here proposed, based on drive-time distance and linear spatial filtering, which allows for reaching a meaningful size in terms of population while still keeping a spatial focus on the central point of the aggregated cluster, i.e., a single municipality as a target, regardless of its dimension.

In accordance with the conducted analysis, we hypothesized that the implementation of data science methods, applied to the georeferenced database of calls and vehicles dispatched by EMS for respiratory and infective causes, could be used to infer early alert monitoring relevant to the COVID-19 spatio-temporal evolution, with anticipation, higher granularity, and more reliability compared to official infection data. In our previous study [34], we already demonstrated the possibility to identify the timing points of change in the shape of the curves of EMS dispatches across different districts, evidencing the possible start of epidemic growth. To further exploit the proposed framework in the direction of providing support for decision-making, in this study our aim was to develop and validate a continuous monitoring model, based on ML methods with supervised learning, for the day-by-day analysis of the evolution of EMS data, in order to generate an ‘early-alert’ COVID-19 system with a higher level of geographic granularity (single municipalities instead of districts with 100,000 residents), to promptly identify the occurrence of new hotspots across the analyzed territory. The main novelties introduced are:The use of a proxy data source—EMS data instead of official swab tests—characterized by a smaller time lag for communication and processing, less dependent on people’s behavior, available infrastructures (also for information flow), and already automatically collected.A high geographic granularity (single municipalities), obtained through spatial processing methods.A simple and agile architecture, both in the data structure and in the computing algorithm, which allows fast execution and daily updates to the model.

The implementation relies on the retrospective data collected by EMS relevant to the Italian region of Lombardy (with a population of 10.06 M inhabitants over a surface of 23,844 km^2^), the first area outside of China to record an outbreak of COVID-19.

## 2. Materials and Methods

### 2.1. Model Development and Optimization

In order to apply supervised ML to tackle a classification problem, the following steps were identified:Definition of the target variable: the class-defining label that the algorithm must assign to each record;Identification of the explicative attributes: measurements on which the classification is based;Identification of the main computational block: ML classification algorithm to be trained and subsequently applied;Definition of a post-processing algorithm, aimed at elaborating the output of the main computational block in order to enhance the representativeness and usability of the output.

To this aim, the methodology and the output of our previously published descriptive model [34] were exploited to define a binary label (step I) corresponding to the active spreading or no diffusion of COVID-19 in a specific territory. More specifically, after dividing the Lombardy region into 77 districts of approximately 100,000 residents, for each district, the time series representing the number of vehicles dispatched by the EMS (normalized by the resident population) for respiratory or infective issues was analyzed to automatically define a first inflection point, representing an estimate of the start of the pandemic spread. This operation was performed using a previously validated signal processing algorithm suited for the identification of inflection points for curves with noise superimposed [47]. Briefly, it is based on a geometrical criterion in which a trapezium is built on the time series and, iteratively, one vertex is moved over time: the position of such a vertex characterized by the maximal trapezoid area is considered as the inflection point. Data preceding this point were labeled as ‘0’ (no diffusion) while data following it were labeled as ‘1’ (active spreading), as represented in Figure 1. To enhance the informative content within the training set, in addition to our previous analysis focused only on the period from 1 January to 23 March 2020, the data relevant to the following waves of the pandemic in Lombardy were also considered. Moreover, the ending points of the waves were also identified by applying the same algorithm to the reversed data. In this way, the final training set was composed of a total of 58,190 daily records, randomized and composed of a balanced share of the two classes, spanning from 1 January 2020 to 13 March 2022 (802 days).

The next step (II) is the identification of the explicative attributes, the features to be computed and associated with the label of each day, on which the algorithm will perform the classification. For this purpose, the series of unfiltered data relevant to both the daily calls to the EMS number and the daily dispatches of EMS vehicles, relevant to respiratory and infective causes, normalized by the resident population in the district under analysis, were considered. This choice was supported by the very high correlation of both the number of ambulances dispatched (r^2^ = 0.81) and emergency calls (r^2^ = 0.96) with official casualties at the province level (the official data with the highest granularity available) due to COVID-19 during the first pandemic peak in Lombardy, as shown in [34].

For each target day, its value and those of the six preceding days were used (7 values for calls + 7 values for ambulances dispatched = 14 attributes), and further elaborated to extract the following features from both signals:‘Position’ features: max value, min value, max-min, time position of the max, and min values in the seven days (5 × 2 = 10 attributes);‘Statistical’ features: mean value, median value, standard deviation (3 × 2 = 6 attributes);Linear regression features: intercept, slope, and Pearson’s correlation coefficient of the linear regression (3 × 2 = 6 attributes);Exponential regression features: base numerical coefficient, exponential coefficient, and Pearson’s correlation coefficient of the exponential regression (3 × 2 = 6 attributes).

The resulting total number of computed features was 42, and a detailed list is provided in Appendix A.

The following step (III) consisted of defining the ML classification algorithm to be trained and consequently applied to classify as ‘0’ or ‘1’ (i.e., ‘no diffusion or ‘active spreading’, respectively) the current target day, based on a retrospective series of 7 data points for each territorial cluster, capturing the possible trends existing in such data, and the relevant label assignment probability. Following a trial-and-evaluation strategy, three different ML approaches (logistic regression, random forest classifier, and support vector machine, widely applied algorithms in the recent literature in the field of EMS demand forecast [48,49,50]), together with different combinations of their explicative attributes, were tested and optimized. Therefore, attribute selection was based on a trial-and-evaluation strategy rather than a priori techniques. The pre-processing of attributes consisted of a single step (the computation of derived attributes), and no further mathematical processing (such as Principal Components Analysis) was implemented.

A performance test was carried out using a 10-fold cross-validation protocol. The whole dataset was balanced in terms of validation labels, with both labels equally represented in terms of the number of records for each district and for each fold in the cross-validation process, thus guaranteeing a balance in both the test and the training set for all iterations; in addition, the order of the records was randomized to avoid feeding the algorithm with almost-periodical cycles.

With the most performant model identified, a further analysis was performed considering the probability of label assignment by the algorithm as a test threshold, thus inferring the area under the curve (AUC) of the receiving operator characteristic (ROC) curves. This analysis was conducted separately for the three main waves of COVID-19 occurring in Lombardy during the whole period under observation (as can be noticed from the example reported in Figure 1):The first wave, in the spring of 2020, with the original strain and no vaccine available.A second wave, from the autumn of 2020 to the spring of 2021, composed of two peaks, the former relevant to the Alpha variant (when vaccinations were not available yet) and the latter to the Delta variant (when vaccinations were available, with an increasing amount of vaccinated people over time).A third wave from the winter of 2021–2022 to the spring of 2022, relevant to the first Omicron variant, despite the high level of vaccination across the population.

To do so, the intervals corresponding to the different waves were computed separately for each district, setting the end of each wave 15 days after the identified ending inflection point. Data from each wave were evaluated as an external dataset, hence excluding them from the training phase, in order to avoid overfitting the model. From these curves, an optimal working point was identified through the index of union [51], allowing us to define an optimized sensitivity and specificity.

### 2.2. Model Post-Processing

Due to the simple and poorly informative binary classification, considering the large variability in the data, and the lack of an actual ground truth to validate the model, the output was further post-processed to be interpreted from a probabilistic perspective (step IV). Specifically, the probabilities associated with the label assignments relevant to four consecutive target days were considered, computing their mean value and mean daily variation between consecutive days, and used separately to define two new features, each characterized by three possible labels:Confidence level: certainly low (mean value < 0.4), uncertain (mean value between 0.4 and 0.6), certainly high (mean value > 0.6);Confidence trend: decreasing (mean variation < −0.1), stable (mean variation between −0.1 and 0.1), increasing (mean variation > 0.1); the +/−0.1 threshold was selected as corresponding to the change in value necessary to move from a fully uncertain confidence level (0.5) to either low or high.

The thresholds for the confidence level were arbitrarily selected, but the reasoning behind this choice was to keep a balance between the post-processing and the original ML output: as a consequence, an uncertainty interval around the original 0.5 threshold was introduced, thus leading to the selected 0.4–0.6 range. The confidence trend thresholds followed consequently.

According to these two features, five ‘alert classes’ were defined as:Class 1: certainly low confidence level with a decreasing or stable confidence trend;Class 2: certainly low confidence level with an increasing confidence trend, or an uncertain confidence level with a decreasing confidence trend;Class 3: uncertain confidence level with a stable confidence trend;Class 4: uncertain confidence level with an increasing confidence trend, or a certainly high confidence level with a decreasing confidence trend;Class 5: certainly high confidence level with a stable or increasing confidence trend.

The number of classes was set to five in order to keep it minimal, targeting the best possible explicability while still accounting for both confidence level and confidence trend. A graphical representation of this post-processing is reported in Figure 2.

### 2.3. Geographical Processing

In order to avoid edge effects and to enhance real-world applicability by considering administrative boundaries, while the model was trained on the same geographical subdivision used in [34], it was instead applied at the level of each single municipality. To do so, it was necessary to compute ‘dynamic’ districts large enough to be statistically meaningful but centered on each single municipality. The first step was to compute driving-time distances among all municipalities, in order to perform an aggregation that could be more representative of a real-world scenario. Specific processing was applied to the city of Milan, which was subdivided into its 9 administrative sub-municipalities. The whole process resulted in a total number of 1514 municipalities in the Lombardy region, for which a 1514 × 1514 driving-time distance matrix was computed. A target of 100,000 residents was set, where a population’s municipalities except the target one (i.e., the center of the district) were weighted according to their distance (with distances scaled in the 0–1 range, and therefore applied as a weight): consequently, the overall absolute population of ‘dynamic’ districts actually resulted higher than 100,000. As a result, the ‘dynamic’ district, centered on a target municipality $\bar{m}$, was obtained by aggregating the closest c=1…N municipalities, with priority given by the driving-time distance $d_{\bar{m}c}$ (scaled between 0 and 1 on the series of all $d_{\bar{m}c}$ with *c* = 1 … 1513) until the weighted total population wtp reached 100,000, with wtp computed as:(1)$$wtp=p_{\bar{m}}+\sum_{1}^{N}p_{c}\times {d_{\bar{m}c}|}_{0-1}\;$$
where $p_{\bar{m}}$ is the resident population in the target (center) municipality $\bar{m}$, $p_{c}$ is the population of municipality c, and ${d_{\bar{m}c}|}_{0-1}$ is the driving-time distance between municipality $\bar{m}$ and municipality c (scaled between 0 and 1 in the entire region).

In order to focus on the target municipality, it was also necessary to differently weight the events (i.e., the dispatch of an ambulance or a call received by the EMS department for respiratory and infective issues) occurring in the ‘dynamic’ district. To this aim, the previously described weighting system was applied, based on the driving-time distance between the municipality where the event occurred ($c_{e}$) and the target municipality $\bar{m}$, thus computing a time series (for each day i) of events as:(2)$$TS_{i}=\sum_{1}^{E_{\bar{m}}}\frac{1}{wtp}+\sum_{1}^{E}\frac{1\times {d_{\bar{m}c_{e}}|}_{0-1}}{wtp}\;$$
with:

$E_{\bar{m}}$ = total number of events during day i in the target municipality $\bar{m}$;

E = total number of events during day i in all the other municipalities within the ‘dynamic’ district;

${d_{\bar{m}c_{e}}|}_{0-1}$ = drive-time distance (scaled between 0 and 1) between the municipality where the event occurred, $c_{e}$, and the target municipality $\bar{m}$.

## 3. Results

### 3.1. Model Development and Optimization

The optimization of the ML algorithm was performed on the basis of the results of the 10-fold cross-validation protocol, separately for each combination of explicative attributes and for each computational algorithm. Three different metrics (precision, recall, F1 score) were considered separately for each of the two labels (i.e., ‘no diffusion’ or ‘active spreading’) and also for all records together (with both flat and weighted averages), also adding the F1 score for accuracy. For all metrics, their distribution in the 10-fold cross-validation process (median, first quartile, third quartile, lower and upper 95% confidence interval) was considered, for a total of 3 × 4 × 5 + 1 × 5 = 65 parameters. However, these 65 parameters were weighted differently in order to give priority to the identification of true positives, thus resulting in a unique final score, representing the evaluation on which the classification decision was based. The different weights applied are reported in Table 2, and the complete results relevant to the final score obtained for each algorithm and tested attribute combinations are reported in Table 3.

According to this strategy, the model that resulted as the most performant was a random forest classifier fed with the ‘position’ and ‘statistical’ attributes of the two signals, yet without the time series, achieving an 81% F1 score (76% sensitivity, 87% specificity) for the ‘no diffusion’ label and a 79% F1 score (73% sensitivity, 85% specificity) for the ‘active spreading‘ label, with a weighted overall score of 0.8052. Noticeably, the range of final scores reached by the different combinations is not wide and multiple combinations resulted in similar values (e.g., 10 combinations resulted in a weighted final score between 0.8 and 0.8052).

As regards the AUC evaluation for the three main waves of COVID-19, the results showed a good performance, with AUC values above 0.87 in the first and third wave, and a slightly lower value of 0.84 for the second wave, as can be expected considering that it was characterized by two distinct peaks and by an extended phase of plateau. The complete results are reported in Figure 3.

### 3.2. Model Post-Processing

As regards the significance of the classification process defined in the post-processing, there was no ground truth available to quantitatively validate its performance. However, it was hypothesized that the correct assignment to a certain class should be reflected by the difference in the number of ambulances dispatched by EMS in the following days. Therefore, this parameter was measured (considering the following 7 days) for all dynamic districts across the whole time series, and value distributions were computed for all five classes. This hypothesis was confirmed by a non-parametric Friedman test among all distributions, which resulted in a *p*-value < 0.001, followed by pairwise Wilcoxon’s rank-sum tests (with Bonferroni correction) that showed p-values largely below 0.05. The complete results of these tests are reported in Table 4.

Within the GIS environment, it was possible to graphically represent the results of the model, thus enabling a powerful and quick visualization and interpretation of the data through mapping. Some examples are reported in the following Figure 4, relevant to the peaks of the three main waves (two different peaks were observed during the second wave) occurring in Lombardy during the analysis period.

## 4. Discussion

In this study, the development and validation of a monitoring model based on supervised learning (ML methods) for the evolution of COVID-19 was proposed, in order to make a step ahead from only descriptive analysis and thus generate an ‘early-alert’ system, with geographic granularity up to single municipalities. The choice of the best-performing algorithm and of the most informative set of attributes was carried out considering the output of the spatial-temporal model presented in [34], and it is therefore based on the optimization of the accuracy in the binary classification. The best results were obtained with a random forest classifier, fed with the ‘position’ and ‘statistical’ attributes of the two signals, yet without the time series, reaching an overall accuracy of 80% (as computed with a 10-fold cross-validation protocol). Therefore, the first relevant result is represented by the higher performance achieved by the random forest classifier when compared with logistic regression and a support vector machine. A possible explanation could be related to the different nature of such ML algorithms: while logistic regression is a mathematical interpolation, and a support vector machine is a separation-based method, the random forest classifier is rule-based. As in our framework, the problem is posed as a classification task, rather than a numerical regression, it is not surprising that the best results are obtained using rule-based reasoning, especially as a consequence of the priority given to the identification of true positives (through the weighting system described in Table 2).

Similar considerations could be drawn for the optimization of the attributes set. The fact that the best results were obtained with ‘position’ and ‘statistical’ attributes can be interpreted as a consequence of the rule-based nature of the algorithm. The numerical series, together with their interpolations, are of low interest for the algorithm, which does not process them as signals but instead makes better use of their derived characteristics. Removing the time series from the input probably frees some computational capability (required to extract the relevant characteristic of the signal), which is redirected towards a better performance in the actual classification. The fact that the support vector machine and logistic regression were achieving better results when considering the linear regression on the signal, along with the statistical parameters, might confirm this hypothesis. It is worth noticing that the results are not largely different across the proposed algorithms, and are even closer across different attribute sets.

As the optimization process can still provide enhancements to the performance, it is not recommended to spend too much effort in this step, as the explainable and interpretable component of the phenomenon can be extracted quite easily, while the possibility of reaching a significantly higher performance is questionable. In this study, the focus was mainly put on the agility and replicability of the proposed framework. A possible increase in the performance, but at the expense of model explicability, could be achieved by exploring the application of different approaches, such as deep learning techniques, that could constitute the aim of future work.

Separate analyses were run on the three main COVID-19 waves that occurred in Lombardy, showing a higher performance on the first and third one, and slightly lower performance on the second wave, which, however, was characterized by a more complex profile, with two distinct peaks separated by a phase of plateau.

The need for research on the predictive capabilities of EMS calls in terms of hospitalizations, ICU demand, and casualties was encouraged in the recent literature [35], and multiple studies were, in fact, published about the use of EMS data in the monitoring and management of COVID-19 emergencies, such as:In October 2020, a study [36] was aimed at comparing trends of EMS data with the official data of COVID-19 cases and related casualties in the area of Tijuana, Mexico. The analysis was focused on two main targets: changes in out-of-hospital mortality, and a comparison of pre- and post-epidemic distributions of the values of oxygen saturation in hospitalized patients. The correspondence between the peaks of the analyzed indicators led the authors to the conclusion that EMS data are a valuable source to monitor excess out-of-hospital mortality due to COVID-19.In November 2020, a retrospective study [37] on the Ile-de-France region, France, was conducted, analyzing the correlation of six healthcare-related parameters, including the number of calls to EMS, with the demand for ICU beds, resulting in a significant time-dependent reproduction ratio with relevance to EMS calls, identifying it as a potentially useful predictor for monitoring and in organizational models to anticipate the demand for ICU beds.In February 2021, a study [38], further elaborating previous results [39] relevant to Kings County, WA (USA), correlated the number of COVID-19 diagnoses in a hospital setting with the identification of suspected patients, considering shortness of breath, cough, sore throat, muscle aches, loss of sense of smell or taste, or diarrhea. A significant correlation was identified, with a peak when considering in the model a nine-day lag period, suggesting that EMS data could anticipate the demand for hospital services, thus confirming the potentiality of such data in the planning of resource allocation and in the management of the healthcare system.A reverse perspective was recently proposed [40] in September 2021, to determine how the number of patients hospitalized for COVID-19 could help in foreseeing the demand for EMS, with reference to the Austin-Travis county (Texas, USA). The authors applied the ‘change point detection’ method to identify changes in the mean and variance of time series, subsequently studying with a *t*-test the distributions in the pre- and post-pandemic periods (as divided by the identified change point). On this basis, a regression model fed with forecasts of COVID-19 hospitalizations was developed and described as a successful method to predict the demand for EMS services, thus further confirming the correlation between these two measurements.

Compared to these state-of-art [36,37,38,39,40] studies, the main strength of the proposed model stands in combining a sound performance with a very high geographic granularity, which reached the level of single municipalities thanks to the proposed combination of ML and spatial filtering, a novel approach in the context of EMS data analysis. This enhancement could be extremely valuable in terms of applicability as support for decision-making, enabling policymakers to differentiate the interventions across the territory rather than managing uniformly the whole area of competence. A first immediate consequence could be hypothesized in the allocation strategy of emergency resources (mainly ambulances and related personnel), which can be optimized on the basis of a detailed demand analysis, thus avoiding a uniform distribution that could result in being under- or over-dimensioned at the same time in different areas, depending on the specific time-bounded needs for each territory.

Moreover, the model stands on minimum requirements in terms of data and processing capabilities. While it can be assumed that environmental, epidemiological, socio-economic, and demographic factors could improve the predictive capabilities of the model, the inclusion of such diverse data sources would pose severe barriers to its replicability and extendibility. By limiting the analysis to a single, widespread, and very simple data source, our model can be very easily implemented by other institutions in different territories.

Performing the validation of the model post-processing results, the main limitation that needed to be faced was the absence of an actual ground truth on which data could be validated, and on which the post-processing could be programmatically trained and calibrated. However, the meaningfulness of the proposed five classes could still be assessed, to some extent, by analyzing the number of vehicles dispatched by the EMS department for respiratory and infective issues in the seven following days, according to the alert class assigned. Indeed, the distributions of the values gave different results, with a progressively increasing number of ambulances dispatched according to the assigned model class, and those differences resulted as statistically significant (see Table 4), thus proving that the proposed strategy to define different alert classes was (at least to some extent) representative of different situations of the near-future evolution of COVID-19. The choice to focus the validation on the demand for EMS was enforced by the priority given to the early identification of an increase in the request for medical assistance, which is of higher concern compared to the bare increase in overall infections, including asymptomatic and mild cases.

From the presented results, some qualitative analysis can be drawn. First of all, the first wave (March–April 2020) appeared to be the worst, with a larger and more synchronous spreading across the territory. However, we cannot explain if this depended more on an effective wider diffusion of the disease, or rather on the management difficulties generated by an unprecedented, unexpected, and abrupt emergency, with a burden on EMS departments beyond the maximal capabilities of the system [46,52]. The second wave (while the official numbers in terms of infections were almost ten times higher than in the first one) appeared close to the first but was less impactful, with a tighter maximal territorial diffusion. Interestingly, a specific geographic area was spared according to the model: it corresponds to a territory located in the provinces of Bergamo and Brescia, which were the most-affected worldwide areas during the first wave [46]. This result might therefore be explained as being due to a stronger natural immunization of the resident population, derived from the previous extreme diffusion of the disease, and to the reduced number of subjects at risk (due to the high mortality during the first peak). Similar considerations can be drawn for the second peak in the second wave, where again the spatial diffusion never covered the entire territory. The third wave appeared very similar to the second phase of the previous one, both qualitatively (in terms of geographic areas) and quantitatively (in terms of diffusion on the territory). It is noteworthy that, during the third wave, a five-fold increase in the daily infection cases with respect to the second one was officially recorded, and yet the impact on EMS was (as modeled in this study) somehow similar, showing how different the impact was on the healthcare system thanks to widespread vaccination. Since the summer of 2021, when vaccine coverage reached significant values in the area, the proposed model is probably more representative of the demand for medical care rather than of the actual diffusion of the disease—two different aspects previously intertwined but currently uncoupled. This characteristic may be valuable in the upcoming future, when it will be vital to suddenly detect any possible surge in the demand for EMS services due to either a new COVID-19 variant, a fall in vaccine-induced immunization, or a combination of these two factors.

## 5. Conclusions

In conclusion, the implementation of a data science approach to infer monitoring information relevant to the evolution of the COVID-19 pandemic, based on the georeferenced database of calls to the emergency number and ambulance dispatched for respiratory and infective issues by EMS, can be considered successful.

In particular, the novel geo-processing algorithm to build ‘dynamic’ weighted districts allowed the ability to reach a very high level of granularity (single municipalities), which is, to the extent of our knowledge, unprecedented for pandemic monitoring models. Compared to official data, the proposed model could be capable of anticipating the detection of new hotspots, thanks to the immediate usability of EMS data, which are (in most cases) automatically collected by EMS organizations and do not require additional structures for information flow (compared, for example, to the swab tests, which require processing, verification, and communication between different actors). Although a direct comparison with official data was not possible due to the absence of an actual ground truth, the proposed model could be considered less biased and more representative of the spread of the disease (at least before the availability of vaccines) and of the demand for medical care (also in the current scenario with a high share of population covered by vaccination).

In this consideration, the developed early alert model could show its high potential in detecting local abrupt surges in COVID-19 in the current situation (July 2022), where periodical recrudescence could be expected in the following months due to the lifting of protection rules and restrictions, in combination with a decline in the vaccine-induced immune response, as well as in the context of possible future pandemics representing unfortunate yet realistic scenarios we must cope with.

## Figures and Tables

**Figure 1 ijerph-19-09012-f001:**
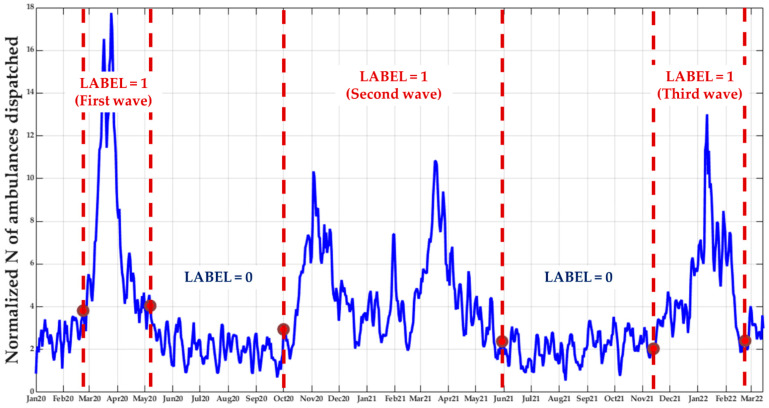
Subdivision of the time series, representing the vehicles dispatched by EMS for respiratory and infective issues (normalized by the resident population) in a certain territory, in periods where a label (0 = ‘no diffusion’; 1 = ‘active spreading’) was assigned to each point generated, according to the automated identification of inflection points (change in the shape of the data, see [34] for more details).

**Figure 2 ijerph-19-09012-f002:**
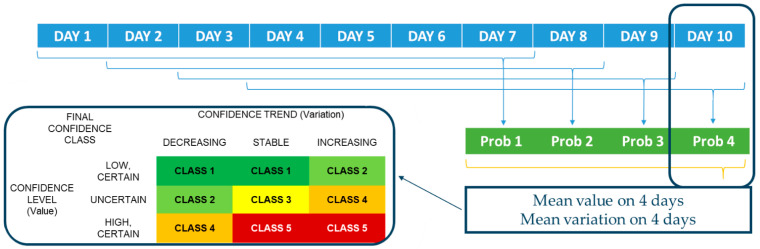
Post-processing elaboration of a machine learning model probability output, applied to classify the time series of ambulances dispatched and calls received by EMS for respiratory and infective issues (see text for details), to distinguish between a scenario of no epidemic diffusion and a scenario of active spreading by generating five possible classes of alert.

**Figure 3 ijerph-19-09012-f003:**
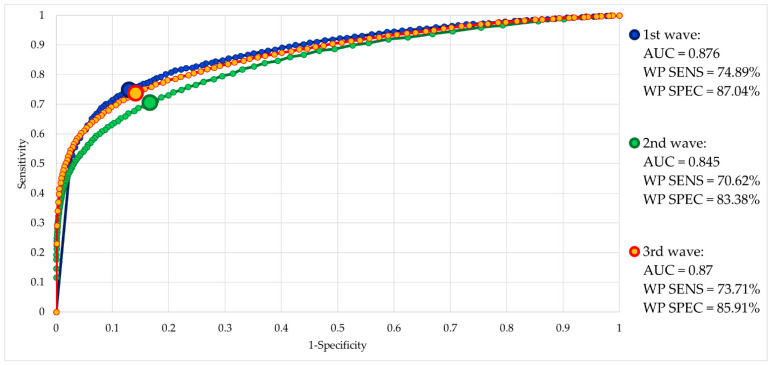
ROC curves representing the performance of the selected machine learning algorithm (random forest classifier) for a daily binary classification of territorial districts in the condition of the active spreading of the COVID-19 epidemic (‘1’) or no diffusion of the epidemic (‘0’) on the basis of ambulance dispatches and calls received by EMS department in Lombardy, Italy, between 1 January 2020 and 13 March 2022. The considered threshold is the label probability as provided in the output by the algorithm. The three COVID-19 waves (spring of 2020, autumn of 2020 to spring of 2021, winter of 2021–2022 to spring of 2022) were evaluated separately, training the model with the data from the other two. The area under the curve was computed, along with the sensitivity and specificity of their optimized [51] working points.

**Figure 4 ijerph-19-09012-f004:**
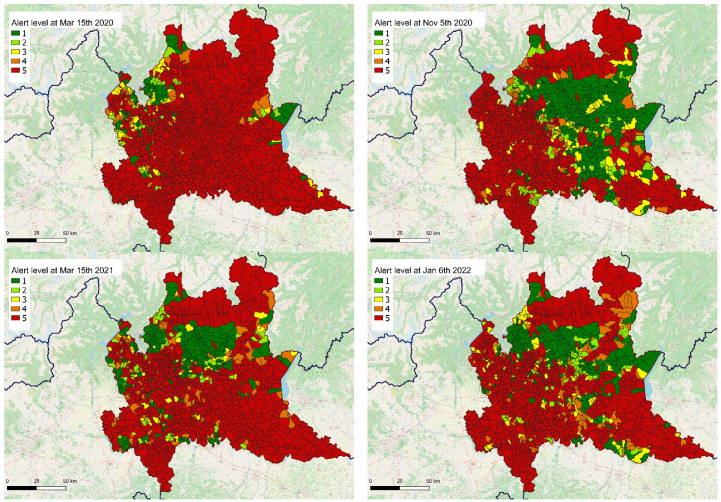
Map representation of the output of a machine learning model representing, for every municipality of Lombardy, Italy, the alert level of confidence, as expressed in five classes, of being in a condition of the active spreading of COVID-19, on the basis of a post-processing of the machine learning output. Four dates are here considered: 15 March 2020 (**top left**) as the peak of the first wave; 5 November 2020 (**top right**), as the first peak of the second wave; 15 March 2021 (**bottom left**), as the second peak of the second wave; and 6 January 2022 (**bottom right**) as the peak of the third wave.

**Table 2 ijerph-19-09012-t002:** Weights applied for the different metrics (precision, recall, F1 score), computed for the ‘no diffusion’ label, ‘active spreading‘ label, and for the whole dataset, to evaluate the performance of a machine learning algorithm trained to identify these two scenarios, as resulting from a 10-fold cross-validation protocol; the median, first quartile, third quartile, and lower and upper 95% confidence interval values across the distribution among the 10 folds are reported (i.e., the median value of recall for the ‘active spreading’ label was weighted 0.1).

Weights Assigned to Different Parameters in the 10-Fold Cross-Validation		Precision	Recall	F1 Score
‘No diffusion’ label	Median	0.03	0.06	0.03
	1st quartile	0.0075	0.015	0.0075
	3rd quartile	0.0075	0.015	0.0075
	95% lower C.I.	0.0075	0.015	0.0075
	95% upper C.I.	0.0075	0.015	0.0075
‘Active spreading’ label	Median	0.05	0.1	0.05
	1st quartile	0.0125	0.025	0.0125
	3rd quartile	0.0125	0.025	0.0125
	95% lower C.I.	0.0125	0.025	0.0125
	95% upper C.I.	0.0125	0.025	0.0125
Accuracy	Median	NA	NA	0.06
	1st quartile			0.015
	3rd quartile			0.015
	95% lower C.I.			0.015
	95% upper C.I.			0.015
Macro Average	Median	0.015	0.03	0.015
	1st quartile	0.0038	0.0075	0.0038
	3rd quartile	0.0038	0.0075	0.0038
	95% lower C.I.	0.0038	0.0075	0.0038
	95% upper C.I.	0.0038	0.0075	0.0038
Weighted Average	Median	0.015	0.03	0.015
	1st quartile	0.0038	0.0075	0.0038
	3rd quartile	0.0038	0.0075	0.0038
	95% lower C.I.	0.0038	0.0075	0.0038
	95% upper C.I.	0.0038	0.0075	0.0038

**Table 3 ijerph-19-09012-t003:** Optimization of a machine learning algorithm for a daily binary classification of territorial districts in the condition of the active spreading of the COVID-19 epidemic (‘1’) or no diffusion of the epidemic (‘0’) on the basis of ambulances dispatched and calls received by the EMS department in Lombardy, Italy, between 1 January 2020 and 13 March 2022. The table reports the final scores of different combinations of machine learning algorithms and attribute combinations (see Appendix A for a detailed list), computed by averaging the different metrics (and their distributions indicators) on the 10-fold cross-validation protocol, according to the defined weights (see Table 2). In bold, the highest results reached for each algorithm are highlighted, while the overall best result is also underlined.

Machine Learning Algorithm: Attributes Included *	Features Numbers (Ref. to Appendix A)	Random Forest	Support Vector Machine	Logistic Regression
All	1–42	0.7967	0.7809	0.7829
Time-Series (TS)	1–14	0.7887	0.7805	0.7818
All Derived Attributes	15–42	0.799	0.7804	0.7826
Ambulances Dispatches	1–7, 15–28	0.7965	0.7806	0.7827
Emergency Calls	8–14, 29–42	0.7939	0.7792	0.7811
Max-Min + TS	1–14, 15–19, 29–33	0.7934	0.7792	0.7819
Max-Min	15–19, 29–33	0.7981	0.7786	0.7798
Statistics + TS	1–14, 20–22, 34–36	0.7975	0.7787	0.78
Statistics	20–22, 34–36	0.8017	0.7791	0.7804
Position and Statistics + TS	1–14, 15–22, 29–36	0.8017	0.7791	0.7805
**Position and Statistics**	15–22, 29–36	** 0.8052 **	0.779	0.7806
Lin Regression + TS	1–14, 23–25, 37–39	0.8039	0.7789	0.7815
Lin Regression	23–25, 37–39	0.8032	0.7792	0.7815
Exp Regression + TS	1–14, 26–28, 40–42	0.8015	0.7597	0.7481
Exp Regression	26–28, 40–42	0.7996	0.7601	0.7482
Lin & Exp Regression	23–28, 37–42	0.7993	0.7604	0.7483
Position + Lin Reg + TS	1–19, 23–25, 29–33, 37–39	0.7983	0.7605	0.7484
Position + Lin Reg	15–19, 23–25, 29–33, 37–39	0.7994	0.7605	0.7484
Position + Exp Reg + TS	1–19, 26–33, 40–42	0.799	0.7605	0.7484
Position + Exp Reg	15–19, 26–33, 40–42	0.7996	0.7605	0.7483
Position + Lin & Exp Reg + TS	1–19, 23–33, 37–42	0.799	0.7606	0.7483
Position + Lin & Exp Reg	15–19, 23–33, 37–42	0.7991	0.7608	0.7484
Statistics + Lin Reg + TS	1–14, 20–25, 34–39	0.7883	0.7799	0.7837
**Statistics + Lin Reg**	20–25, 34–39	0.7974	**0.** **7815**	**0.7841**
Statistics + Exp Reg + TS	1–14, 20–22, 26–28, 34–36, 40–42	0.8005	0.761	0.749
Statistics + Exp Reg	20–22, 26–28, 34–36, 40–42	0.8009	0.7611	0.7489
Statistics + Lin & Exp Reg + TS	1–14, 20–28, 34–42	0.8002	0.7611	0.7501
Statistics + Lin & Exp Reg	20–28, 34–42	0.8003	0.7611	0.7503

* LEGEND (see Appendix A for detailed list): TS (time series) = daily calls to the EMS number, daily dispatches of EMS vehicles, relevant to respiratory and infective causes, normalized by the resident population; POSITION = max value, min value, max-min, position of the max and min values in the time window; STATISTICS = mean, median, standard deviation; LIN REG (Linear regression) intercept, slope, and Pearson’s correlation; EXP REG (Exponential regression): base numerical coefficient, exp coefficient, and Pearson’s correlation.

**Table 4 ijerph-19-09012-t004:** Median (25th–75th percentile) of the number of ambulances dispatched for respiratory and infective issues by the EMS department on the territory of Lombardy, Italy, relevant to the 5 classes representing the level of confidence of being in a situation of the active spread of COVID-19, assigned by post-processing from the machine learning algorithm output for each municipality, between 1 January 2020 and 13 March 2022 (see text for details); the last column reports the *p*-values of pairwise Wilcoxon’s rank-sum tests (after Bonferroni correction), assessing the difference in the distribution across different classes.

Assigned Class	Ambulances Dispatched/Population in the Following 7 Days: Median [25th–75th Percentile]	*p*-Value of Pairwise Wilcoxon’s Rank-Sum Tests (Bonferroni Corrected)
Class 1	1.83 [0.96–2.55]	Class 2: *p* < 0.001 Class 3: *p* < 0.001 Class 4: *p* < 0.001 Class 5: *p* < 0.001
Class 2	3.21 [2.57–4.16]	Class 1: *p* < 0.001 Class 3: *p* < 0.001 Class 4: *p* < 0.001 Class 5: *p* < 0.001
Class 3	3.73 [2.85–4.69]	Class 1: *p* < 0.001 Class 2: *p* < 0.001 Class 4: *p* < 0.001 Class 5: *p* < 0.001
Class 4	3.96 [3.00–5.03]	Class 1: *p* < 0.001 Class 2: *p* < 0.001 Class 3: *p* < 0.001 Class 5: *p* < 0.001
Class 5	6.24 [4.63–9.00]	Class 1: *p* < 0.001 Class 2: *p* < 0.001 Class 3: *p* < 0.001 Class 4: *p* < 0.001

## Data Availability

All data are available on reasonable request.

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
