# Peer review of "Generating High-Granularity COVID-19 Territorial Early Alerts Using Emergency Medical Services and Machine Learning"

_ijerph, 2022, doi:10.3390/ijerph19159012_

Round 1

Reviewer 1 Report

[Comment 1] Novelty

[Subcomment 1a] (lines 95-98) The novelty of this study is unclear. The authors need to list similar and related studies, then compare them with this study to show the novelty. I suggest the authors present the comparison in a table and list the contributions clearly in Section 1.

[Subcomment 1b] (lines 115-117) To be clear with the novelty of this study, please show what is new in this study in contrast to [27] at the end of Section 1.

[Comment 2] Methods and experiments

[Subcomment 2a] (lines 119-120) How do the authors differentiate umber of ambulances dispatched for COVID or any other disease? Also, how do the authors set the threshold to differentiate label 1 and 0 (please state the rule clearly).

[Subcomment 2b] (lines 138-140) Please state any reference that justify the consideration of the number of calls and number of dispatches for predicting the disease spread. There should be an appropriate reasoning here.

[Subcomment 2c] (line 153) For clarity, I suggest the authors to add a list of the 42 features in the appendix.

[Subcomment 2d] (lines 159-160) Please add references showing that the selected ML techniques have good performance when dealing with similar cases considered in this study.

[Subcomment 2e] (Section 2.2) How did the authors set the threshold values? Please list the reference or state an appropriate reasoning.

[Subcomment 2f] What do the authors suggest based on Table 2, in terms of ML model and data selection? How would the authors justify that they suggested a good method, when the results are (1) similar between many methods) and (2) contradicting for the STATISTICS + LIN REG data (very bad when using random forest, but good when using others). I could not see any clear conclusion for now.

[Comment 3] Analysis

I believe the authors need to discuss more about the ML methods in Section 4, because the main part of this study is the ML models. Please add at least one or a half page analysis about the model selection, and possibly the data selection as well.

[Comment 4] Writing quality and clarity

[Subcomment 4a] (lines 66-69) Please rephrase the sentences to allow a better explanation flow considering the previous and next paragraphs.

[Subcomment 4b] (lines 161-163) The sentence cannot be understood easily. Please provide a further explanation. I would think that the statement is incorrect because even when using a preprocessed signal data, we can still use the principal component analysis.

[Subcomment 4c] (lines 175-181) Could you please relate this information with Figure 1. Presenting Figure 1 that covers these time intervals would be a better example.

[Subcomment 4d] (Left bottom part of Figure 2 and lines 202-209) Instead of presenting the information using table rows, it would be understood better when using a matrix form with confidence levels at the row title and confidence trend at the column title. Please revise Figure 2.

[Subcomment 4e] It is difficult to understand Table 1. Please add an example on how to read a cell, e..g, accuracy with F1 Score, then active spreading label with recall (in the text).

[Subcomment 4f] Please be consistent in giving the underline or not in Table 2.

[Subcomment 4g] Please revise the numbering errors in lines 350, 357, and 363.

Reviewer 2 Report

This paper put forward COVID-19 early-alert system based on Emergency Medical Services (EMS) data. It is a reasonable work because the EMS data indeed reveal the epidemic status.   

1) As shown in table 2. POSITION AND STATISTICS are only a little bit better than LIN REGRESSION + TS and LIN REGRESSION. It seems hard to find one certain kind of parameters or a combination from table 2. I suggest authors try to explain why POSITION AND STATISTICS could achieve the best results.

2) add the details about features, and how to get them.

3) it is hard to understand table 1.

Reviewer 3 Report

A scientifically interesting hypothesis, correctly posed and verified. An important goal of research of a clear utilitarian nature and of great social importance. A well-chosen, classic Covid-19 region with a sufficiently large database for analysis. Research procedure well presented. Methods and tools correctly selected for the planned research.

It can be argued whether the adopted value "> 0.6" is appropriate for the term certainly high (line: 197), in my opinion the ranges should be slightly different: (from 0.0 to 0.3; from> 0.3 to 0 , 7; from> 0.7 to 1.0), but this is the sovereign decision of the Authors.

It is worth emphasizing the geo-processing algorithm, which allowed the analyzed data to achieve a high level of detail (at the level of municipalities), which may be crucial for the emergence of the next wave of Covid-19.

Generally, I consider the work assessed as good. However, I believe that in order to emphasize the originality of the adopted solution, the literature (part of the Introduction) should be expanded to include works in the field of Covid-19 modeling and forecasting, such as, for example:

Alsayed, A .; Sadir, H .; Kamil, R .; Sari, H. Prediction of Epidemic Peak and Infected Cases for COVID-19 Disease in Malaysia, 2020. Int. J. Environ. Res. Public Health 2020, 17, 4076.

Hussein, T .; Hammad, M.H .; Surakha, O .; AlKhanafseh, M .; Fung, P.L .; Zaidan, M.A .; Wraith, D .; Ershaidat, N. Short-Term and Long-Term COVID-19 Pandemic Forecasting Revisited with the Emergence of OMICRON Variant in Jordan. Vaccines 2022, 10, 569.

Lynch C., Gore R. Application of one-, three-, and seven-day forecasts during early onset on the COVID-19 epidemic dataset using moving average, autoregressive, autoregressive moving average, autoregressive integrated moving average, and naïve forecasting methods. Data in Brief 2021;35:106759

Lynch C., Gore R. Short-Range Forecasting of COVID-19 During Early Onset at County, Health District, and State Geographic Levels Using Seven Methods: Comparative Forecasting Study, J Med Internet Res 2021;23(3):e24925

Wynants, L .; Van Calster, B .; Collins, G.S .; Riley, R.D .; Heinze, G .; Schuit, E .; Bonten, M.M.J .; Dahly, D.L .; Damen, J.A .; Debray, T.P.A .; et al. Prediction models for diagnosis and prognosis of covid-19: Systematic review and critical appraisal. BMJ 2020, 369, m1328.

Reviewer 4 Report

The paper was relatively well-written and addresses an important problem. However, the contribution should be clarified.

In conclusion, it says that  "the rtPCR swab tests, which require 24-48 hours for processing", which is not true across the globe. Also, there are rapid anibody tests available. It is unclear how this methods is superior to antibody tests.

It also says " a direct comparison with official data was not possble". Therefore, it is unclear the validity of the proposed method.

I would expect the vaccination rate environmental factors (e.g., temperature), and characteristics of the prevailing strain of the virus play important roles in the prediction model. However, I did not see them in the model.

Also, there could be other causes of increase in EMS calls related to respiratory issues such as flu, which might be affected by flu seasons. How does the model account for it?

A set of 5 classes are considered for different geographical areas. I would expect a poorer performance as compared to a set of 2 classes. SOme more discussion on the choices of 5 classes are needed.

A careful proofreading or professional editing is needed. There are some minor grammar errors. E.g.,

p.1, ln.26: "effective in"->"in effectively".

p.2,ln74: "one only"->"only one"; "a high"->"high".

p.3, ln.128: "composed by"->"composed of".

p.4,ln.138: "to this purpose"->"For this purpose".

p.2,ln.66-69: please avoid single sentence paragraphs. Please reorganize the texts. 

Round 2

Reviewer 1 Report

The authors need to add their study into Table 1.

Author Response

We thank the Reviewer for this observation. Acordingly, our study was added as last line in Table 1.

Reviewer 2 Report

The authors have addressed all my comments. I recommend to accept this paper.

Author Response

We thank the Reviewer for his/her appreciation of our work.

Reviewer 4 Report

I applaud the authors for addressing my comments.

Here are some minor comments. Thanks!

p.8, ln. 283: "figure"->"Figure".

p.9, ln. 338, 339; p.11, ln. 364; p.15, ln.435: "table"->"Table".

Author Response

We thank the Reviewer for his/her appreciation of our work.

Minor comments were taken into account, as well as English was revised throughout the manuscript.